# Response to First-Line Treatment with Immune-Checkpoint Inhibitors in Patients with Advanced Cutaneous Squamous Cell Carcinoma: A Multicenter, Retrospective Analysis from the German ADOReg Registry

**DOI:** 10.3390/cancers14225543

**Published:** 2022-11-11

**Authors:** Maximilian Haist, Henner Stege, Berenice Mareen Lang, Aikaterini Tsochataridou, Martin Salzmann, Peter Mohr, Dirk Schadendorf, Selma Ugurel, Jan-Malte Placke, Michael Weichenthal, Ralf Gutzmer, Ulrike Leiter, Martin Kaatz, Sebastian Haferkamp, Carola Berking, Markus Heppt, Barbara Tschechne, Patrick Schummer, Christoffer Gebhardt, Stephan Grabbe, Carmen Loquai

**Affiliations:** 1Department of Dermatology, University Medical Center Mainz, 55131 Mainz, Germany; 2Department of Pathology, Stanford University School of Medicine, Stanford, CA 94305, USA; 3Department of Microbiology & Immunology, Stanford University School of Medicine, Stanford, CA 94305, USA; 4Department of Dermatology and National Center for Tumor Diseases, University Hospital Heidelberg, 69120 Heidelberg, Germany; 5Department of Dermatology, Elbe Kliniken Buxtehude, 21614 Buxtehude, Germany; 6Department of Dermatology, University Hospital Essen, 45122 Essen, Germany; 7Department of Dermatology, Campus Kiel, University Hospital of Schleswig-Holstein Hospital, 24105 Kiel, Germany; 8Department of Dermatology and Allergy, Skin Cancer Center Hannover, 30625 Hannover, Germany; 9Center of Dermatooncology, Department of Dermatology, Eberhard-Karls-University, 72076 Tuebingen, Germany; 10Department of Dermatology, Wald-Klinikum Gera, 07548 Gera, Germany; 11Department of Dermatology, University Hospital Regensburg, 93053 Regensburg, Germany; 12Department of Dermatology, Uniklinikum Erlangen, CCC Erlangen-EMN, Friedrich-Alexander University Erlangen-Nürnberg, 91054 Erlangen, Germany; 13KRH Klinikum Neustadt am Rübenberge, Specialist for Internal Medicine Hematology and Oncology, 31535 Neustadt, Germany; 14Department of Dermatology, University Hospital Würzburg, 97080 Würzburg, Germany; 15Skin Cancer Center, Department of Dermatology and Venereology, University Hospital Hamburg-Eppendorf (UKE), 20246 Hamburg, Germany; 16Department of Dermatology, Gesundheit-Nord Hospital Bremen, 28102 Bremen, Germany

**Keywords:** advanced cutaneous squamous cell carcinoma, checkpoint inhibitor therapy, cemiplimab, immunosuppression, response durability, real-world data

## Abstract

**Simple Summary:**

Cutaneous squamous cell carcinoma (cSCC) is one of the most common malignancies of the skin with poor survival outcomes in advanced stages of the disease. Recent clinical trials demonstrated the efficacy of checkpoint-inhibitors (CPI) therapy for advanced stage disease, but there is a lack of data from real-world cohorts and trial-ineligible patients. In this retrospective, real-world cohort study, we investigated the efficacy of first-line checkpoint-inhibitor treatment in 39 patients with advanced cSCC from eight different German cancer centers and stratified outcomes by the immune status of the patients. Our data demonstrate that patients receiving CPI achieved high response rates with durable remissions in about 20% of patients. CPI also evoked tumor responses in patients with active autoimmune diseases and lymphoproliferative disorders, although these responses were often short-lived, resulting in a significantly shorter overall survival. Notably, CPI therapy was safe with only 15% of patients discontinuing for toxicity.

**Abstract:**

Cutaneous squamous cell carcinoma (cSCC) is a common malignancy of the skin and has an overall favorable outcome, except for patients with an advanced stage of the disease. The efficacy of checkpoint inhibitors (CPI) for advanced cSCC has been demonstrated in recent clinical studies, but data from real-world cohorts and trial-ineligible cSCC patients are limited. We retrospectively investigated patients with advanced cSCC who have been treated with CPI in a first-line setting at eight German skin cancer centers registered within the multicenter registry ADOReg. Clinical outcome parameters including response, progression-free (PFS) and overall survival (OS), time-to-next-treatment (TTNT), and toxicity were analyzed and have been stratified by the individual immune status. Among 39 evaluable patients, the tumor response rate (rwTRR) was 48.6%, the median PFS was 29.0 months, and the median OS was not reached. In addition, 9 patients showed an impaired immune status due to immunosuppressive medication or hematological diseases. Our data demonstrated that CPI also evoked tumor responses among immunocompromised patients (rwTRR: 48.1 vs. 50.0%), although these responses less often resulted in durable remissions. In line with this, the median PFS (11 vs. 40 months, *p* = 0.059), TTNT (12 months vs. NR, *p* = 0.016), and OS (29 months vs. NR, *p* < 0.001) were significantly shorter for this patient cohort. CPI therapy was well tolerated in both subcohorts with 15% discontinuing therapy due to toxicity. Our real-world data show that first-line CPI therapy produced strong and durable responses among patients with advanced cSCC. Immunocompromised patients were less likely to achieve long-term benefit from anti-PD1 treatment, despite similar tumor response rates.

## 1. Introduction

Cutaneous squamous cell carcinoma (cSCC) has the second highest incidence among skin cancers [1], with increasing rates in aging western populations [2]. While the majority of cSCCs are cured with surgery, approximately 5% of patients develop advanced disease that is defined either by locoregional or distant metastases or locally advanced cSCC not amenable to surgery or radiotherapy [3]. These patients have poor long-term outcomes as treatment options in the pre-checkpoint inhibitor (CPI) era were largely limited to palliative chemotherapy [4]. While these treatments have anti-tumor efficacy, responses are often short-lived with significant concomitant toxicity [3,5].

The initial success of CPI-blocking programmed cell death protein 1 (PD-1) or PD-1 ligand (PD-L1) to treat metastatic melanoma also gave an incentive for application in nonmelanoma skin cancers. As cSCC is a highly immunogenic tumor that features high somatic mutational burden, there was a strong rationale for treatment with CPI [3,6]. In accordance with this, data from phase I/ II clinical trials [7,8] demonstrated that advanced cSCC patients who were treated with the PD-1 inhibitors cemiplimab or pembrolizumab [9,10] achieved an objective response rate between 40 and 50% with >50% of these responses lasting longer than 6 months.

Despite the evidence reported by clinical trials, only limited data are thus far available about CPI activity in real-world cohorts [4,6,11,12,13] and several questions still need to be addressed, such as safety and efficacy in patients usually excluded from clinical trials and potential determinants of clinical benefit to CPI treatment [14]. In this regard, patients with chronic immune suppression (i.e., high doses of corticosteroids for autoimmune diseases or organ recipients) and concomitant immunocompromising hematological diseases are of particular interest.

In this study, we, therefore, investigated patients treated with first-line CPI for advanced cSCC outside of clinical trials at eight German skin cancer centers and stratified outcomes by the immune status of the patients.

## 2. Materials and Methods

### 2.1. Study Design and Data Source

In this retrospective, multicenter study, we used the data of eligible patients from the skin cancer registry of the ADOReg [15]. The ADOReg platform collects healthcare data on skin cancer patients from 59 skin cancer centers, eight of which contributed to the current study (Buxtehude, Erlangen, Regensburg, Neustadt, Mainz, Würzburg, Hamburg, and Gera). Details on treatment and outcome specifics were recorded in an unidentifiable, pseudonymized form at the patient level.

### 2.2. Patient Cohort

At data request (February 2022), 436 patients with cSCC were identified within the ADOReg database. Thereof, we analyzed 39 patients with locally advanced, regionally or distant metastatic, or inoperable cSCC who received at least one dose of CPI agents in a first-line setting between February 2018 and June 2022 with follow-up until data cut-off in July 2022 (Figure 1). CPI agents included cemiplimab, nivolumab, pembrolizumab, and avelumab.

The collected data comprised core patient and tumor characteristics (i.e., age, gender, comorbidities, and immune status), sites of metastasis, LDH serum levels, as well as treatment specifics and survival outcomes. Immunocompromised patients either received immunosuppressive medication (chronic steroid use >10 mg prednisolone/day) or were diagnosed with hematological malignancies. The primary endpoint of this study was OS. Secondary endpoints included PFS, real-world tumor response (rwTR), and severe treatment-related adverse events (trAE) as defined in Appendix A. Time-to-next treatment (TTNT) was included as an additional outcome parameter due to its role as a reliable surrogate for OS in real-world datasets [16].

### 2.3. Statistical Analysis

Descriptive statistics were used to analyze baseline characteristics. Treatment duration was calculated as the period between initial drug administration and treatment discontinuation. The chi-square test was used to assess the association between immune status and tumor response rates. For categorial variables, 95% confidence intervals (CIs) were calculated using the Clopper–Pearson method. Testing for equality between subgroups was performed using Student’s t-test and Fisher’s exact test.

We employed Kaplan–Meier survival plots to illustrate median OS and PFS probabilities and to explore associations between the immune status and survival outcomes. Survival curves were compared using the log-rank test. The median duration of follow-up was calculated using the reverse Kaplan–Meier method. Cox’s proportional hazards models were applied to identify predictors of patient survival by adjusting for baseline characteristics, treatment regimen, and immune status. Multivariable analysis was conducted for significant variables by the univariate test or by a priori selection for biological relevance to evaluate their conjoint, independent effects on PFS or OS. In all cases, two-tailed *p*-values were calculated and considered significant with values *p* < 0.05. SPSS version 27, RStudio (Version 1.3.1093), and GraphPad PRISM version 5 were used for all analyses.

## 3. Results

### 3.1. Baseline Patient Characteristics

We evaluated a total of 39 patients who received first-line CPI for advanced cSCC. The median follow-up upon CPI initiation was 27 months (Table 1). Patients were predominantly male and of advanced age (median: 78 years). Therapies prior to receiving CPI included surgery (n = 29/39; 74.3%) or radiotherapy (74.3%). Seven patients were considered inoperable due to extensive disease or field cancerization. Among all 39 patients, nine patients had a history of immunosuppression due to hematological malignancies (chronic lymphocytic leukemia, n = 4; polycythemia vera, n = 2) or immunosuppressive medications for autoimmune diseases (Crohn´s disease, n = 1; autoimmune hepatitis, n = 1; Lichen planus, n = 1; Table 2).

Primary cSCC tumors were mainly located in the head/neck area (64.1%). Most patients initially presented with advanced stage III or IV disease (51.3%) or locally advanced tumors (vertical tumor thickness >6 mm, poorly differentiated histology, or horizontal diameter >2 cm). A total of 30 patients later developed regional (53.8%) or distant metastases (23.1%), while seven patients showed locally advanced tumors that required treatment with CPI.

All patients received at least one dose of CPI, which included cemiplimab (48.7%), nivolumab (25.6%), pembrolizumab (23.1%), or avelumab (2.4%). The median treatment duration was 5.0 months with nine patients continuing treatment at data cut-off. RwTRR was 48.6%, with 10 patients achieving PR and 7 showing CR. Response rates were similar regardless of CPI used (*p* = 0.768). Treatment-related AE were reported for 34.3% of patients with 15.4% of patients ceasing CPI therapy due to toxicity. Following CPI discontinuation due to progression or intolerance, 10 patients (25.6%) received subsequent systemic therapies, which included EGFR inhibitors (5.1%), chemotherapy (2.6%), or CPI re-challenge (20.5%).

The median PFS among the entire cohort was 29.0 months and the median TTNT as well as median OS were not reached at data cut-off in July 2022.

### 3.2. Clinical and Pathological Factors Associated with Response and Survival upon CPI Therapy

When analyzing the association between baseline clinical factors and survival outcomes, univariate Cox-regression analysis showed that longer CPI treatment duration, response to CPI therapy, and good performance status were associated with a longer PFS (Appendix A). In line with this, patients with good performance status showed favorable response to CPI therapy (*p* = 0.008). In addition, the response to first-line CPI treatment and the absence of immunosuppressive medical conditions were correlated with OS (Appendix A). While LDH serum levels correlated with response to CPI therapy (*p* = 0.03, Appendix A), we found no significant association with survival outcomes.

Given the number of target events and the biological rationale, we included the immune status, performance status, rwTR, and treatment duration in a multivariable Cox regression analysis model (Figure 2). In this multivariable analysis, the presence of immunosuppression (HR: 11.8; *p* = 0.053) and the best response to CPI (HR: 0.09; *p* = 0.064) were associated with OS, while only the best tumor response (HR: 0.23; *p* = 0.006) was significantly associated with PFS.

Results from Kaplan–Meier analysis for the variables affecting PFS and OS are shown in Appendix A.

### 3.3. Response and Survival upon CPI Therapy in Immunocompromised Patients

We next investigated the impact of the immune status on response to first-line CPI therapy. It is well known that immunocompromised patients are at higher risk of developing locoregional or distant metastases and that immunosuppression is an adverse prognostic factor in advanced cSCC. However, our analysis unveiled no significant differences in tumor response of immunocompromised patients as compared to immunocompetent patients (*p* = 0.093). In addition, we found no significant differences for rwTRR (48.1% vs. 50.0%, *p* = 1.0) or rwTCR (85.2% vs. 62.5%, *p* = 0.321) (Appendix A and Appendix A).

Regarding survival outcomes, our analysis revealed that immunocompromised patients had a significantly shorter median OS (29 months vs. NR, *p* < 0.001) and TTNT (12 months vs. NR; *p* = 0.016) as compared to immunocompetent patients. Our data also showed that patients with immunosuppressive conditions presented with a shorter PFS (11 vs. 40 months, *p* = 0.059), although this association was below statistical significance (Figure 3).

### 3.4. Duration of CPI Treatment Response in Immunocompromised Patients

To explain the divergent survival outcomes for immunocompromised patients, we analyzed the treatment outcomes of responders to CPI therapy. The median follow-up time for this patient cohort was 31 months (95% CI: 23.4–38.6 months). During this follow-up period, we observed that 2/4 patients with baseline immunosuppression progressed at 3 and 11 months of follow-up, while among immunocompetent responders, 10/13 patients (76.9%) remained relapse-free. In line with this, immunocompromised patients who showed at least PR to CPI therapy had a significantly shorter median PFS (11 vs. 40 months, *p* = 0.0046) compared to immunocompetent responders, suggesting that tumor responses in this patient subgroup less frequently result in durable tumor remissions (Figure 4 and Figure 5). Among the 17 responders, 16 were alive at data collection.

### 3.5. Durable Response upon CPI Cessation and Efficacy of CPI Re-Challenge

Thirteen patients discontinued CPI therapy during ongoing remission (7 PR, 6 CR, or no evidence of disease, NED) at a median of 9 months. Reasons for treatment cessation included CPI-associated toxicities (pneumonitis, grade 2; hepatitis, grade 3; fatigue), the explicit wish of the patient, or cessation-sustained tumor remission. Within the subsequent 22-month follow-up-period, three patients relapsed. Notably, among all patients who discontinued CPI in ongoing remission, three patients had an immunosuppressive condition and 2 of them relapsed shortly after treatment cessation (median: 1.5 months). Among these patients, one patient with concomitant B-CLL was re-challenged with cemiplimab after relapse of the primary tumor in the genital area and achieved stable disease thereafter.

Overall, 10 patients received subsequent treatments upon disease progression after initial CPI treatment. Among these patients, 7 were re-challenged with CPI (1 CR; 2 PR; 3 SD; 1 not assessed, due to early discontinuation for severe AE), 1 was treated with taxol-based chemotherapy (SD), 1 received EGFR-inhibitor cetuximab (PR), and 1 patient received a combination of cemiplimab and cetuximab (not assessed at data cut-off).

### 3.6. Treatment-Related Adverse Events during CPI Therapy

Among the entire cohort, seven patients developed AE **of** CTCAE grade III or higher (17.9%), which included hepatitis (n = 2), colitis (n = 2), pancytopenia (n = 1), fatigue (n = 1), anaphylactic shock (n = 1), and pancreatitis (n = 1). Thereof, three patients permanently discontinued CPI therapy. Serious AE were more frequent among immunocompromised patients, although this association was below statistical significance (*p* = 0.319, Table 1). Among patients with serious AE, six achieved at least stable disease upon CPI therapy with three of these showing an ongoing tumor remission. Other documented side-effects of grade II included diarrhea, colitis, exanthema, increased liver enzymes, pyrexia, fatigue, thyroiditis, and pneumonitis. Upon CPI re-challenge, 3/7 patients developed trAE, including exanthema, elevated liver enzymes, pneumonitis, and colitis.

## 4. Discussion

In this study, we provided real-world data on the efficacy and safety in a well-defined cohort of advanced cSCC patients with extended follow-up times who received first-line CPI therapy, which was complemented by data on survival outcomes for immunocompromised patients. Our data confirmed efficacy results for first-line CPI therapy from previous clinical trials, transferring them into real-world cases. In particular, our data on rwTRR (48.5%) and rwTCR (76.4%) are consistent with response rates reported earlier [7,8,17]. Notably, we observed a higher rate of patients achieving CR to CPI treatment (20.0%), which might be attributed to the longer follow-up period of our trial, the fact that our patients were treated with CPI in a first-line setting, and a potential sampling bias due to the smaller sample size.

Further, and in line with previous reports analyzing the durability of CPI responses for melanoma, we observed that CPI evoked durable responses, particularly among immunocompetent patients that showed at least PR. These tumor responses were ongoing even after CPI-cessation. Of note, patients who restarted CPI after previously having achieved at least SD upon CPI therapy also showed at least SD upon CPI rechallenge. Next, we observed that an impaired performance status, weak response to CPI, and a shorter treatment duration were associated with shorter survival outcomes. Contrasting previous reports, we could not detect a higher probability of treatment response for patients with the primary site of the head-neck area [6,12]. Further, we observed a significant correlation between elevated serum LDH levels and a weaker response to CPI therapy, as previously described for melanoma [18] and more recently for advanced cSCC [6].

Most importantly, our real-world data allowed us to better define the efficacy and safety of CPI therapy in a subgroup of immunocompromised patients. These patients are at higher risk of developing cSCC and present with a more aggressive course of the disease [19]. As immunocompromised patients were largely excluded from clinical trials investigating CPI efficacy, evidence for these patients is limited to small observational studies with heterogeneous cohorts. Thus far, data indicated that patients with concomitant hematological diseases achieved lower response rates and shorter PFS-periods upon CPI therapy [20] and that CPI were associated with a higher risk of severe AE [12,21]. By contrast, our data demonstrate that patients with active autoimmune disease and lymphoproliferative disorders achieved comparable response rates as compared to immunocompetent patients, highlighting the feasibility and activity of CPI in patients with such immunosuppressive conditions, as reported in previous series [12,20]. However, further analysis also showed that these patients were less likely to achieve durable remissions and disease progression was frequently observed within our follow-up period, resulting in a substantially shorter PFS for this subcohort.

In our cohort of advanced cSCC patients, we found low rates of severe trAE, with less than 20% discontinuing CPI therapy due to toxicity, which is in line with previous reports for CPI safety in advanced melanoma [22,23]. Given the higher rate of severe AE in immunocompromised patients, which could be explained by the immunological imbalance inherent to the patients´ autoimmune disorder, and the potential development of life-threatening events, we propose a consistent monitoring of AE during and after CPI therapy.

We acknowledge the limitations of our study, including the retrospective nature, which adds a selection bias. When discussing safety data of this real-world population, the nonstandardized documentation of safety data should be considered, which might have affected the identification of AE, particularly those of lower grade, resulting in the underrepresentation of AE. Moreover, the heterogeneity in terms of disease status, autoimmune diseases, and medication might have affected our results. Most importantly, the small number of patients with autoimmune diseases and concomitant immunosuppressive treatments must be considered a relevant limitation.

In summary, we provide real-life data from a multicenter analysis that confirms the safety and efficacy of first-line CPI therapy in advanced cSCC patients. Among the entire cohort, 48.5% of patients achieved a tumor response to first-line CPI with a median PFS of 29.0 months. Although our study was exploratory in its nature and, therefore, does not allow for definitive conclusions, our data indicate that immunocompromised patients were able to achieve similar response rates without significantly increased toxicities. Therefore, CPI therapy may offer a promising treatment approach for these high-risk cSCC patients. Given that remissions are often short-lived in this patient cohort with substantially shorter PFS and that severe AE may occur at any time during treatment, an individual approach to therapy and consistent monitoring will be necessary for these patients.

## 5. Conclusions

Our retrospective, multicenter analysis demonstrates that first-line CPI therapy evokes strong and durable tumor responses in a real-world cohort of advanced cSCC patients. Importantly, our data provide evidence that patients with immunosuppressive conditions, such as active autoimmune diseases or lymphoproliferative disorders, show similar response rates but are less likely to achieve durable tumor remissions and frequently develop tumor recurrence.

## Figures and Tables

**Figure 1 cancers-14-05543-f001:**
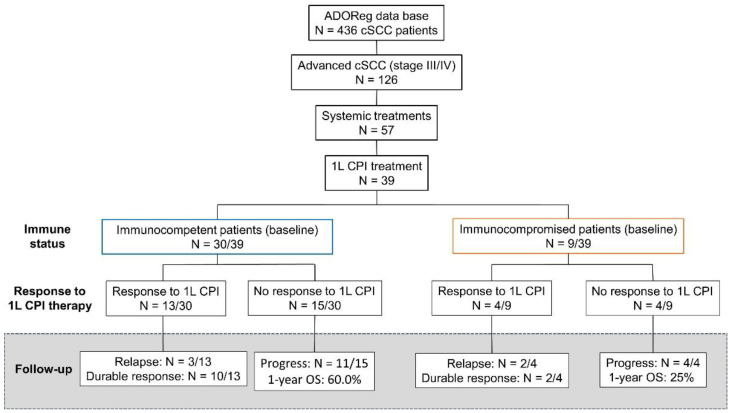
Flow chart depicting the selection criteria for this retrospective, multicenter analysis. We analyzed the outcome of patients with advanced cutaneous squamous cell carcinoma (cSCC) who were treated with first-line (1L) immune checkpoint inhibitors (CPI) at stratified outcomes based on the immune status at baseline. Patients without significant immunosuppression generally showed more durable tumor responses compared to immunocompromised patients despite similar rates of initial tumor responses. In the case of 3 patients with ongoing CPI therapy (2 among immunocompetent patients and 1 patient with baseline immunosuppression), no tumor assessments were available at the time of data cut-off.

**Figure 2 cancers-14-05543-f002:**
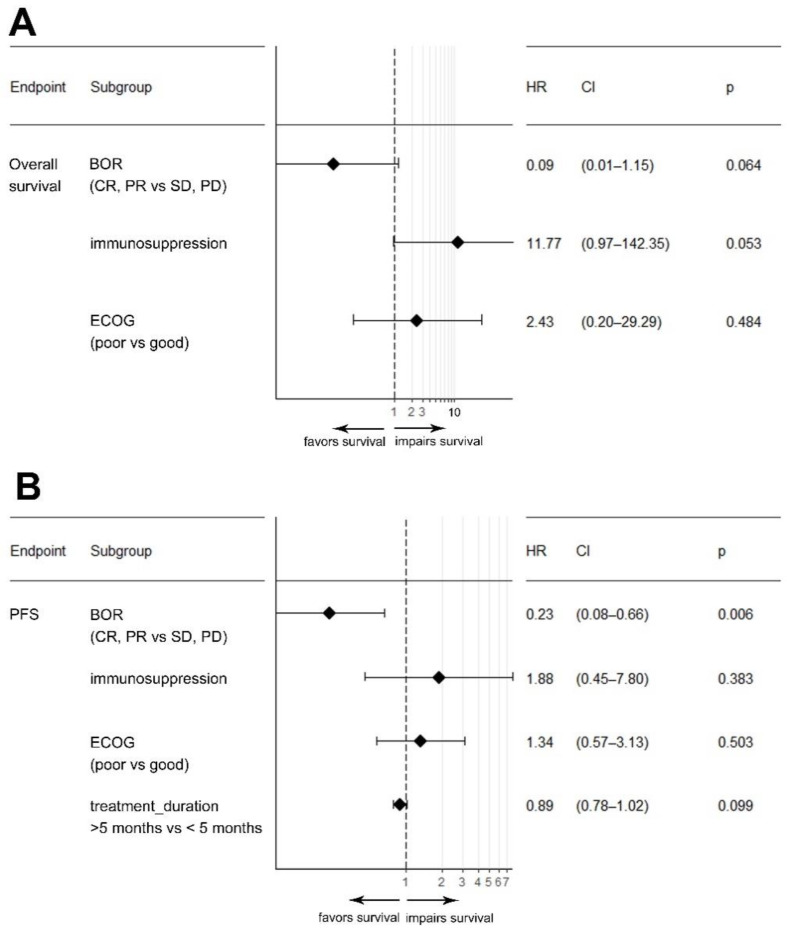
Forest plot depicting results from a multivariable Cox-regression model for the variables real-world tumor response, immunosuppression, and ECOG performance status on overall survival (**A**) and progression-free survival (PFS) (**B**). Abbreviations: BOR: best overall response; CR: complete response; CI: confidence interval; ECOG: eastern cooperative oncology group; HR: hazard ratio; PR: partial response; PD: progressive disease; SD: stable disease.

**Figure 3 cancers-14-05543-f003:**
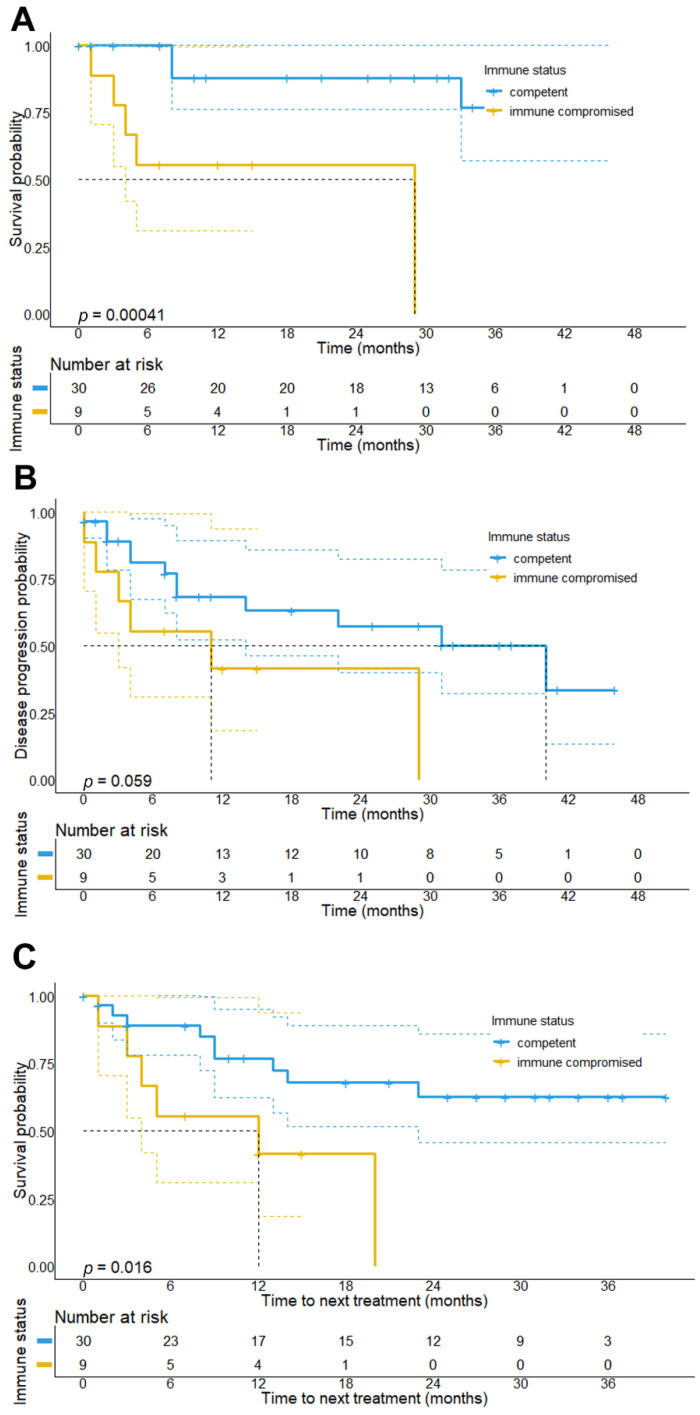
Overall survival, progression-free survival, and time-to-next treatment (TTNT) stratified by the immune status of the patients. Results show that patients being immunocompromised at the start of CPI treatment have a significantly shorter overall survival (29 months vs. NR, *p* < 0.001) (**A**) and progression-free survival (11 months, 95% CI: 0–28.1 months vs. 40 months, 95% CI: 16.9–63.1 months, *p* = 0.059) (**B**), as well as TTNT (median TTNT: 12 months, 95% CI: 0–29.1 months vs. NR; *p* = 0.016) (**C**) as compared to immunocompetent patients.

**Figure 4 cancers-14-05543-f004:**
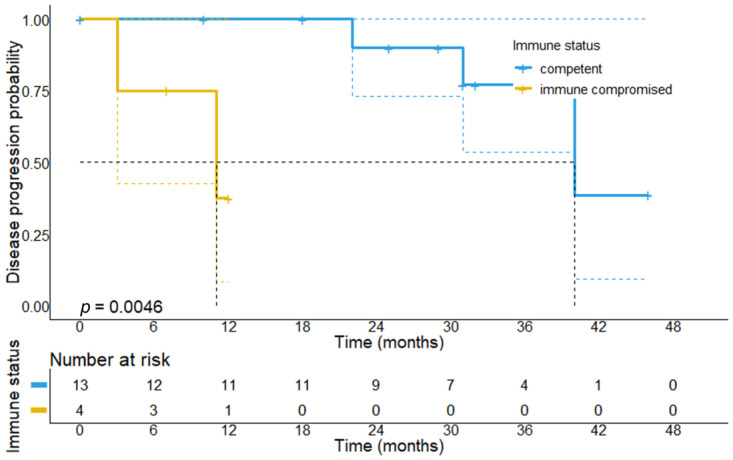
Progression-free survival in patients who achieved response to first-line CPI treatment stratified by the immune status. It can be found that immunocompromised responders have a significantly shorter PFS (11 months, 95% CI: 0–23.0 months vs. 40 months, 95% CI: 27.1–52.9 months, *p* = 0.0046) as compared to immunocompetent patients, which suggests that responses in immunocompromised patients are short-lived, whereas immunocompetent patients achieve more durable responses.

**Figure 5 cancers-14-05543-f005:**
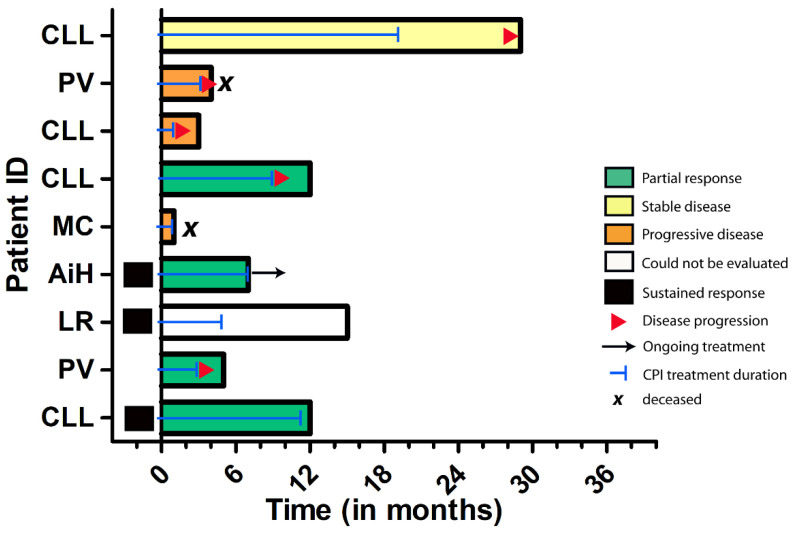
Swimmer’s plot depicting the individual outcomes of immunocompromised patients in this study. Among those patients receiving first-line CPI treatment, three patients showed sustained tumor responses, whereas the majority of patients experienced tumor progression in the course of the disease. Abbreviations: CLL = chronic lymphocytic leukaemia, PV = polycythemia vera; MC = Crohn´s disease, AiH = autoimmune hepatitis, LR = lichen planus mucosae.

**Table 1 cancers-14-05543-t001:** Baseline patient characteristics and treatment outcomes.

Clinicopathological Characteristics	Overall Patient Cohort	Immunocompetent Patients	Immunocompromised Patients	*p* Value
Number of patients	39	30	9	
** *A Demographics* **				
Age at CPI initiation (median, range)	79 years (55–96)	79.5 years (55–96)	79.0 years (61–87)	0.812
*Gender*				0.607
Female	6 (15.4%)	4 (13.3%)	2 (22.2%)	
Male	33 (84.6%)	26 (86.7%)	7 (77.8%)	
*Median ECOG performance status* ^1^				0.062
Good performance status (ECOG ≤ 1)	17 (60.7%)	16 (69.6%)	1 (20%)	
Poor performance status (ECOG > 1)	11 (39.2%)	7 (30.4%)	4 (80%)	
*Primary site of disease*				0.604
Head and neck	25 (64.1%)	19 (63.3%)	6 (66.7%)	
Limb	8 (20.5%)	6 (20.0%)	2 (22.2%)	
Trunk	4 (10.3%)	4 (13.3%)	0	
Genitoanal area	2 (5.1%)	1 (3.3%)	1 (11.1%)	
*Immunosuppression*				NA
None	30 (76.9%)	30	0	
Autoimmune disease	2 (5.1%)	0	2 (22.2%)	
Hematological disease	6 (15.4%)	0	6 (66.7%)	
Other (immunosuppressive medication)	1 (2.6%)	0	1 (11.1%)	
*Initial stage at diagnosis* ^2^				0.451
Stages I and II	17 (43.6%)	15 (50.0%)	2 (22.2%)	
Stages III and IV	20 (51.3%)	14 (46.7%)	6 (66.7%)	
Unknown	2 (5.1%)	1 (3.3%)	1 (11.1%)	
*High-risk features*				0.168
Diameter > 2 cm ^3^	8 (20.5%)	4 (13.3%)	4 (44.4%)	
Vertical thickness > 6 mm ^4^	8 (20.5%)	5 (16.7%)	3 (33.3%)	
Poorly differentiated histology ^5^	12 (30.8%)	11 (36.7%)	1 (11.1%)	
Other pathological risk factors ^6^	5 (12.8%)	5 (16.7%)	0	
Time to metastasis (median, range)	8 months (0–72)	12 months (0–72)	1.0 month (0–19)	**0.023**
*Extent of disease* ^7^				0.518
Locally advanced	7 (17.9%)	7 (23.3%)	0	
Regional metastasis	21 (53.8%)	15 (50.0%)	6 (66.7%)	
Distant metastases	9 (23.1%)	7 (23.3%)	2 (22.2%)	
NA	2 (5.2%)	1 (3.3%)	1 (11.1%)	
*Anatomic sites of metastasis*				0.399
Lymphatic tissue	23 (59.0%)	17 (56.7%)	6 (66.7%)	
Soft tissue/skin	12 (30.8%)	8 (26.7%)	4 (44.4%)	
Bone	4 (10.3%)	3 (10.0%)	1 (11.1%)	
Lung	4 (10.3%)	2 (6.7%)	2 (22.2%)	
(Lepto-)meningeal	3 (7.7%)	3 (10.0%)	0	
Other (mucosal, intraorbital)	3 (7.7%)	3 (10.0%)	0	
*Baseline LDH levels* ^8^				1.0
Normal (<245 U/l)	19 (48.7%)	15 (50.0%)	4 (44.4%)	
Elevated (>245 U/l)	15 (38.5%)	12 (40.0%)	3 (33.3%)	
** *B Treatments* **				
*Initial treatment regimen for advanced disease*				0.06
Surgery alone	9 (23.1%)	9 (30.0%)	0	
Definitive RTx	9 (23.1%)	8 (26.7%)	1 (11.1%)	
Surgery + RTx	20 (51.3%)	12 (40.0%)	8 (88.9%)	
None	1 (2.6%)	1 (3.3%)	0	
*CPI regimens*				0.387
Nivolumab	10 (25.6%)	8 (26.7%)	2 (22.2%)	
Pembrolizumab	9 (23.1%)	7 (23.3%)	2 (22.2%)	
Avelumab	1 (2.6%)	0	1 (11.1%)	
Cemiplimab	19 (48.7%)	15 (50.0%)	4 (44.4%)	
*Real-world tumor responses*				0.093
Progressive disease, PD	7 (20.0%)	4 (14.8%)	3 (37.5%)	
Stable disease, SD	11 (28.9%)	10 (37.0%)	1 (12.5%)	
Partial response, PR	10 (28.5%)	6 (22.2%)	4 (50%)	
Complete response, CR	7 (20.0%)	7 (25.9%)	0	
Not assessed ^9^	4	3	1	
Median duration of CPI treatment	5 months (0–29)	4.5 months (0–29)	5.0 months (0–19)	0.98
Treatment-related adverse events	12 (34.3%)	9 (30.0%)	4 (44.4%)	0.689
Serious adverse events	7 (17.9%)	4 (13.3%)	3 (33.3%)	0.319
Discontinuation due to trAE	7 (15.4%)	4 (13.3%)	2 (22.2%)	0.653
*Subsequent treatments*				1.0
Re-induction of CPI therapy	8 (20.5%)	7 (23.3%)	1 (11.1%)	
EGFR-inhibitor	2	0	2	
Chemotherapy	1	1	0	
** *C Survival outcomes* **				
Median overall survival (95% CI)	Not reached	Not reached	29.0 months	**<0.001**
1-year OS	32 (82.1%)	27 (90%)	5 (55.5%)	
Deceased	9 (23.1%)	4 (13.3%)	5 (55.6%)	**0.018**
Median progression-free survival (95% CI)	29.0 months (8.6–49.4)	40.0 months (16.9–63.1)	11.0 months (0–28.1)	0.059
Disease progression or relapse	18 (46.2%)	12 (40%)	6 (66.7%)	0.255
Median follow-up period (95% CI)	27.0 months (21.7–32.2)	29.0 months (24.0–34.0)	12.0 months (6.6–17.4)	0.242

Abbreviations: trAE: treatment-related adverse events; CI: confidence interval; CPI: checkpoint-inhibitors; CR: complete response; ECOG: Eastern Cooperative Oncology Group; EGFR: epidermal-growth factor receptor; NR: not reached; OS: overall survival; PD: progressive disease; PR: partial response; RTx: radiotherapy; SD: stable disease. ^1^ ECOG performance status was unknown for 11 patients; ^2^ numbers apply for all known AJCC stages at initial diagnosis (n = 37); ^3,4,5^ statistics apply for all patients with known horizontal tumor diameter (n = 10), vertical tumor thickness (n = 28), and pathological grading (n = 29); ^6^ other risk factors include: sarcomatoid-like transformation, desmoplasia, perineural invasion, lymph vessel infiltration, or osseous infiltration; ^7^ advanced tumor stage was unknown for 2 patients; ^8^ statistics apply for patients with known LDH levels in serum at baseline (n = 34); ^9^ no documentation of tumor response due to preliminary treatment cessation.

**Table 2 cancers-14-05543-t002:** Baseline patient characteristics and treatment outcomes for patients diagnosed with autoimmune diseases.

AID	AID-Status at CPI Start	Concomitant Medication 30 d Prior to CPI Initiation	Steroids during CPI	AE during CPI Therapy	AE Resolved?	Permanent CPI Cessation?	BOR to CPI
Crohn´s disease	Acute relapse	Mesalazine 2 g/d Prednisolone > 10 mg/d	No	Anaphylactic shock	Yes	Yes	PD
AiH	Active	Prednisolone 10 mg/d Azathioprine 1 mg/kg bw	Yes	Pancytopenia grade 3	Yes	No	PR
Lichen planus mucosae	Active	Prednisolone >10 mg/d Acitretin 0.5 mg/kg bw Topical Betamethasone	No	Pneumonitis grade 2	Yes	Yes	PR

Abbreviations: AE: adverse events; AID: autoimmune disease; AiH: autoimmune hepatitis; BOR: best overall response; bw: body weight; CPI: checkpoint inhibitors; PD: progressive disease; PR: partial response.

## Data Availability

All relevant data are within the manuscript and its Appendix A. The retrospective data used for statistics have been collected within the framework of the ADOReg.

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
