# Peer review of "Response to First-Line Treatment with Immune-Checkpoint Inhibitors in Patients with Advanced Cutaneous Squamous Cell Carcinoma: A Multicenter, Retrospective Analysis from the German ADOReg Registry"

_cancers, 2022, doi:10.3390/cancers14225543_

Round 1

Reviewer 1 Report

Dear editor, 

thank you for sending me this manuscript for review 

this is an interesting study with  important implications in  our daily practice in the management of  advanced cutaneous squamous cell  carcinoma. 

Except from some English errors that needs to be corrected, i don't have any additional comments  

Author Response

We thank the reviewer for the positive feedback on our article and have improved English language and style in the revised version of our manuscript. 

Reviewer 2 Report

This study about the cutaneous squamous cell carcinoma (cSCC) is based on the retrospective analysis of the efficacy of the checkpoint inhibitors (CPI; blocking molecule PD-1) in  39 advanced cSCC patients from eight different cancer centers (remark: a small group of patients for such a number of centers).
Among these patients, nine have been assigned as  immunocompromised (impaired immune status as a result of immunosuppressive medication or hematological diseases). This small group is a very heterogenous in terms of medical status.
The first conclusion that “first-line CPI therapy produced strong and durable responses among patients with advanced cSCC”  is not an innovative observation. The second conclusion that  patients with immunosuppressive conditions (active autoimmune diseases,  lymphoproliferative disorders) show similar response rates but get shorter progression-free survival time, is a worth noticed observation about patients usually excluded from clinical trials investigating CPI efficacy. This observation could be treated as a presumption that needs further observation. 

Author Response

We thank the reviewer for the kind assessment of our research and have taken into account the valuable criticism in our revised manuscript. In particular, we agree with the reviewer that further studies are needed in order to confirm the results on CPI efficacy in patients with baseline immunosuppression. 

Reviewer 3 Report

This current work by Haist et al., validates the efficacy of  immune-checkpoint inhibitors in patients with advanced cutaneous squamous cell carcinoma (cSCC).They found  immune-checkpoint inhibitors with high response rates and with durable remission in 20 % patients. In addition to that checkpoint inhibitors therapy produced strong and durable responses among patients with advanced cSCC. Moreover, they validated their effectiveness in immunocompromised patients and showed its effectiveness. 

 I am in principle supportive of accepting this work for publication.

Author Response

We thank the reviewer for the kind assessment of our research article and the comments.

Reviewer 4 Report

The authors are trying to provide real-life data that confirms the safety and efficacy of first-line CPI therapy in advanced sSCC patients. One of the limitations of this research is the number of patients with autoimmune diseases and concomitant immunosuppressive treatments, maybe request access to more centers data or European centers. 

Something that I found relevant in this study is that they found similar results in line with previous reports that suggest that this study was performed with care and methodologically. 

Author Response

We thank the reviewer for valuable feedback and comments. Indeed, the small number of patients is a relevant limitation of this study and efforts to include data from more skin cancer centers to provide more powerful results are currently underway.
